# OpenReview forum: "A2D: Any-Order, Any-Step Safety Alignment for Diffusion Language Models"
_ICLR.cc/2026/Conference — ICLR 2026 Poster_

### Official Review · Reviewer_SJBR · 2025-10-26

**Soundness:** 3
**Presentation:** 3
**Contribution:** 3
**Rating:** 6
**Confidence:** 4

**Summary:**

This paper proposes A2D (Any-Order, Any-Step Defense), a novel token-level safety alignment method for diffusion large language models (dLLMs).
Unlike autoregressive (AR) models, dLLMs generate tokens in any order, which enlarges the attack surface and makes them susceptible to new types of jailbreaks such as Prefilling and DIJA attacks.
A2D trains dLLMs to emit an [EOS] refusal signal whenever harmful spans appear, by replacing masked harmful tokens with [EOS] during alignment.
This ensures the model can halt generation upon encountering unsafe continuations—at any position and at any decoding step.
Empirically, A2D drastically reduces DIJA success rates (from >80% to ≈0%) across multiple dLLMs (LLaDA, LLaDA-1.5, Dream) while preserving task capability.
Moreover, A2D introduces a real-time safety signal: the probability of [EOS] acts as an online indicator of harmfulness, enabling early rejection (up to 19.3× faster safe termination).
Overall, this work identifies a core safety vulnerability in dLLMs and proposes a simple yet effective solution through token-level alignment.

**Strengths:**

- Introduces the first token-level alignment strategy tailored to dLLMs’ flexible decoding process.

- The method is theoretically consistent with masked diffusion objectives and empirically verified through KL divergence analyses.

- Reduces attack success rates to nearly zero across diverse settings while maintaining comparable MMLU, Math, and Coding performance.

- [EOS] probabilities provide a clear, real-time safety signal enabling early refusal.

- Strong performance across decoding orders, generation lengths, and extreme attacks (FITS).

- The paper explicitly verifies that using [EOS] as the refusal token does not harm normal task performance (see Table 3); in fact, it yields the highest capability scores among alternatives while preserving safety.

- The writing and figures are excellent.

**Weaknesses:**

- While intuitively motivated, the paper lacks a more formal explanation of why [EOS]-based token-level suppression leads to stable refusal boundaries.
- Additional discussion on the training or inference cost overhead of A2D would be helpful.
- Evaluation mainly uses BeaverTails; broader validation across more diverse safety datasets would further strengthen the claims.

**Questions:**

- Could the [EOS]-based refusal mechanism lead to premature refusals in complex, long-form tasks? A quantitative analysis of false rejections beyond XSTest would be valuable.
- Is the early-rejection threshold τ transferable across models, or does it require per-model calibration?
- Does A2D affect long-context coherence or cause early truncation in benign generations?

---

> ### Author Response · Authors · 2025-11-18
> **Rebuttal - Part I**
>
> We thank the reviewer for the encouraging and thoughtful evaluation. We appreciate that the reviewer clearly recognized our core contribution in introducing the first token level safety alignment strategy designed for dLLMs’ flexibility and any-order decoding. We hope the following responses help clarify the remaining points.
>
> ---
>
> **1. Stable Refusal Boundaries**
>
> Thank you for this insightful question. A useful way to interpret A2D is to view the model as learning an implicit binary classifier $q_\text{clf}$ over contexts. For a given partially masked context $X_\text{ctx}$​, the training objective encourages the model to behave as if
>
> > $p_\theta(\texttt{[EOS]} \mid X_{\text{ctx}})\approx q_{\text{clf}}(\text{harmful} \mid X_{\text{ctx}})$
>
> and, for any non-$\texttt{[EOS]}$ token T,
>
> > $p_\theta(T \mid X_{\text{ctx}}) \approx q_{\text{clf}}(\text{not harmful} \mid X_{\text{ctx}})p_\theta^{\text{gen}}(T \mid X_{\text{ctx}}),$
>
> where $p_\theta^{\text{gen}}$ is the model’s standard generative distribution.
>
> Under A2D training, any masked position inside a harmful answer is labeled as $\texttt{[EOS]}$, while masked positions in benign answers are labeled with their true tokens. Because diffusion decoding repeatedly presents many corrupted partial views of the same harmful span, the model learns to make this harmful/not-harmful decision consistently across a wide range of contexts. This repeated supervision across randomized masks and timesteps yields a robust, distributed decision boundary, rather than a brittle dependence on a single token logit.
>
> We have added this classifier-based interpretation and the above factorization to **Appendix C.3** of the revised manuscript to clarify why $\texttt{[EOS]}$-based token-level supervision produces stable and generalizable refusal boundaries.
>
> ---
>
> **2. Clarity on Computational Overhead**
>
> We thank the reviewer for raising this important point. A2D is trained using the standard finetuning procedure for dLLMs and does not introduce any additional computational cost. As shown in the table below, A2D matches the compute cost of other finetuning-based methods such as RT and SFT, when all models are trained under the same LoRA setting, as detailed in **Appendix A**.
>
> * Computational cost (with 64 prompt tokens + 256 response tokens, batch size 1)
>
> ||RT|SFT|VRPO|A2D|
> |-|-|-|-|-|
> |Compute Cost (TFLOPs)|9.7|9.7|29.0|9.7|
>
> During inference, A2D also incurs no extra overhead, since $\texttt{[EOS]}$ probabilities are already computed as part of the standard decoding process. Moreover, the token-level $\texttt{[EOS]}$ signal enables early rejection, resulting in up to 19.3× faster termination on harmful prompts in LLaDA-1.5 (Figure 8). We have added a comprehensive computational cost analysis in **Appendix A.4**.

---

> ### Author Response · Authors · 2025-11-18
> **Rebuttal - Part II**
>
> **3. Diverse Safety Benchmarks**
>
> We agree with the reviewer that expanding the evaluation to more diverse safety benchmarks will strengthen the paper. Following this, we additionally evaluated A2D on AdvBench [1] and JailbreakBench [2], extending the scope of our analysis beyond HarmBench [3], which we mainly evaluated. As shown in the tables below, A2D achieves strong results on both benchmarks and consistently outperforms other finetuning-based methods by a large margin.
>
> Importantly, while A2D is trained solely on BeaverTails [4], it generalizes effectively to previously unseen threat patterns and adversarial structures. This highlights A2D's robustness and generalization to unseen threats. We have added these results in **Section 6.4**.
>
> * LLaDA Results
>
> **AdvBench**
> |Method|Zeroshot ↓|PAIR ↓|ReNe ↓|Perfilling ↓|DIJA ↓|**Avg.** ↓|
> |-|-|-|-|-|-|-|
> |RT|**0.0**|7.5|57.5|82.5|80.0|45.5|
> |SFT|45.0|76.3|63.8|77.5|82.5|69.0|
> |VRPO|0.0|10.0|13.8|1.3|11.3|7.3|
> |A2D|**0.0**|**1.3**|**12.5**|**0.0**|**0.0**|**2.8**|
>
> **JailbreakBench**
> |Method|Zeroshot ↓|PAIR ↓|ReNe ↓|Perfilling ↓|DIJA ↓|**Avg.** ↓|
> |-|-|-|-|-|-|-|
> |RT|2.5|20.0|57.5|70.0|61.3|42.3|
> |SFT|51.3|87.5|55.0|72.3|63.8|66.0|
> |VRPO|**1.3**|21.3|16.3|7.5|22.5|13.8|
> |A2D|**1.3**|**7.5**|**15.0**|**1.3**|**1.3**|**5.3**|
>
> * Dream Results
>
> **AdvBench**
> |Method|Zeroshot ↓|PAIR ↓|ReNe ↓|Perfilling ↓|DIJA ↓|**Avg.** ↓|
> |-|-|-|-|-|-|-|
> |RT|**0.0**|13.8|51.3|33.8|81.3|36.0|
> |SFT|16.3|46.3|40.0|70.0|85.0|51.5|
> |VRPO|**0.0**|10.0|48.8|7.5|60.0|25.3|
> |A2D|**0.0**|**0.0**|**7.5**|**0.0**|**0.0**|**1.5**|
>
> **JailbreakBench**
> |Method|Zeroshot ↓|PAIR ↓|ReNe ↓|Perfilling ↓|DIJA ↓|**Avg.** ↓|
> |-|-|-|-|-|-|-|
> |RT|**0.0**|16.3|41.3|48.8|67.5|34.8|
> |SFT|27.5|67.5|36.3|68.8|76.3|55.3|
> |VRPO|**0.0**|7.5|38.8|18.8|45.0|22.0|
> |A2D|**0.0**|**6.3**|**8.8**|**0.0**|**0.0**|**3.0**|
>
> [1] Zou, Andy, et al. "Universal and transferable adversarial attacks on aligned language models." arXiv preprint arXiv:2307.15043 (2023).\
> [2] Chao, Patrick, et al. "Jailbreakbench: An open robustness benchmark for jailbreaking large language models." Advances in Neural Information Processing Systems 37 (2024).\
> [3] Mazeika, Mantas, et al. "Harmbench: A standardized evaluation framework for automated red teaming and robust refusal." 41st International Conference on Machine Learning (2024).\
> [4] Ji, Jiaming, et al. "Beavertails: Towards improved safety alignment of llm via a human-preference dataset." Advances in Neural Information Processing Systems 36 (2023).

---

> ### Author Response · Authors · 2025-11-18
> **Rebuttal - Part III**
>
> **4. Premature Rejections in Long-Form Tasks**
>
> We thank the reviewer for raising this important concern. To directly assess the risk of false rejections in benign yet complex long-form queries, we constructed a targeted subset from OR-Bench [1], which contains over 80K prompts. Specifically, we selected the 70 longest queries that appear harmful but are in fact safe.
>
> As shown in the table below, A2D achieves the highest compliance rate among all methods on this challenging set. Notably, on LLaDA-1.5, A2D even surpasses the original LLaDA-1.5 model in accurately preserving benign completions without premature refusals. These results suggest that A2D’s refusal mechanism maintains robustness and context sensitivity, avoiding unnecessary truncation even in long-form settings. These results also indicate that A2D is less prone to early truncation on long benign queries. We have included these results in **Section 6.4**.
>
> * Compliance Rate (↑)
>
> |Model|Original|RT|VRPO|A2D|
> |-|-|-|-|-|
> |LLaDA-Instruct|55.7|25.7|44.3|47.1|
> |LLaDA-1.5|45.7|28.6|20.0|54.3|
> |Dream|14.3|4.3|12.9|12.9|
>
> [1] Cui, Justin, et al. "Or-bench: An over-refusal benchmark for large language models." 42nd International Conference on Machine Learning (2025).
>
> ---
>
>
> **5. Transferability of Early-Rejection Threshold ($\tau$)**
>
> A2D’s early-rejection threshold $\tau$ is largely transferable across models, and we observe stable behavior without requiring per-model calibration in practice. As shown in Figure 9 **(Appendix C.2)**, LLaDA, LLaDA-1.5, and Dream all maintain XSTest accuracy above 95% throughout the interval $\tau \in [0.8, 0.999]$, while the early rejection rate increases smoothly with higher thresholds. This consistency indicates that a single global threshold suffices in practice.

---

### Official Review · Reviewer_jYvv · 2025-10-30

**Soundness:** 3
**Presentation:** 2
**Contribution:** 3
**Rating:** 4
**Confidence:** 3

**Summary:**

This paper introduces A2D (Any-Order, Any-Step Defense), a novel token-level safety alignment method specifically designed for diffusion large language models (dLLMs). The authors first identify a critical vulnerability in dLLMs: standard alignment methods are "shallow" and only provide refusal signals at the beginning of a response, making them highly susceptible to template-based prefilling attacks (like DIJA) that inject harmful content at arbitrary positions and steps. A2D addresses this by training the model to predict an [EOS] token (as a universal refusal signal) at any masked position that corresponds to harmful content. This token-level supervision provides robust safety alignment that persists at "any-step" and in "any-order" of the diffusion decoding process.

**Strengths:**

**Novelty and Significance:** The paper tackles a highly significant and novel problem at the frontier of LLM safety. It clearly diagnoses a critical vulnerability specific to the emerging dLLM paradigm (vulnerability to any-order, any-step attacks) and proposes a solution that is directly tailored to that paradigm.

**Effectiveness and Simplicity:** The A2D method is simple and elegant, cleverly repurposing the existing [EOS] token as a distributed refusal signal. The experimental results are extremely strong, showing a near-complete neutralization of powerful, model-specific attacks (DIJA, FITS) against which other alignment methods (RT, SFT, VRPO) fail.

**Weaknesses:**

**Potential for Alignment Tax:** While the paper claims model capability is preserved (Tables 1 & 8), the results are not uniformly dominant. For instance, on LLaDA-1.5, A2D scores slightly lower on the "General" capability mean than RT and lower on "Math" than VRPO. A more in-depth analysis of this potential "alignment tax" on more nuanced or complex benign tasks would be beneficial.

**Limited Scope of Harmful Content:** The alignment dataset is based on BeaverTails. It is unclear how A2D would perform against a wider, more diverse range of harmful content categories (e.g., subtle misinformation, complex reasoning-based harms) that may not be well-represented in this dataset.

**Scalability of Training:** The A2D method requires fine-tuning on a dataset of harmful prompts paired with unsafe responses, where harmful spans are identified and masked. It's not clear how this data-generation and training process scales to more complex, "in-the-wild" harmful requests that may not follow clear templates.

**Questions:**

A2D trains the model to output **[EOS]** for any masked token within a harmful span. How does this "all-or-nothing" token-level refusal handle prompts with mixed intent (i.e., requests that are partially harmful and partially benign)? Does it ever lead to over-refusal, or does it successfully "censor" only the harmful parts while completing the rest?

The "early rejection" mechanism (Section 7) relies on the **[EOS]** probability at the leftmost masked position in the first decoding step. Why was the leftmost position chosen? Given the "any-order" nature of dLLMs, wouldn't an aggregate signal (e.g., the mean or max [EOS] probability across all masked tokens) be a more robust heuristic for refusal?

---

> ### Author Response · Authors · 2025-11-18
> **Rebuttal - Part I**
>
> Thank you for the reviewer’s thoughtful and constructive feedback. Your comments highlighted several important aspects of A2D and helped us strengthen the presentation. We appreciate the careful evaluation and hope that our revisions address the concerns raised.
>
> ---
>
> **1. Alignment Tax**
>
> We thank the reviewer for this important point. While A2D does not always top every capability metric, the differences are negligible in practice. For example, on Math, A2D scores 44.8% vs. 46.0% for VRPO on LLaDA-1.5 and 55.9% vs. 56.2% on Dream. Conversely, on Coding, A2D performs slightly better, achieving 1.5%, 3.5%, and 1.3% higher scores than VRPO on LLaDA, LLaDA-1.5, and Dream, respectively. These variations are well within typical run-to-run fluctuation and do not indicate any meaningful capability tradeoff.
>
> More importantly, these minor variations stand in clear contrast to A2D’s substantial safety gains. A2D reduces DIJA and FITS attack success rates to nearly zero, whereas competing methods remain highly vulnerable (for example, VRPO at 68.1% vs. A2D at 0.0% on Dream). A2D delivers substantial safety gains while keeping the capability differences modest, a result of its token-level alignment mechanism that integrates safety signals deeply yet preserves normal model behavior. We describe this tradeoff in **Lines 332-334**.
>
> ---
>
> **2. Generalization of A2D**
>
> We thank the reviewer for raising this important question regarding the limited scope of harmful content categories in the alignment dataset. While A2D is trained solely on BeaverTails [1], we emphasize that it does not overfit to this specific training distribution. To directly test A2D’s generalization ability, we additionally evaluated it on AdvBench [2] and JailbreakBench [3], which feature threat types that go beyond those in BeaverTails, such as subtle misinformation and complex reasoning-based harms. As shown in the tables below, A2D achieves consistently strong performance across both benchmarks, significantly outperforming other finetuning-based baselines.
>
> Crucially, many of the adversarial templates and harm structures in AdvBench and JailbreakBench do not appear in BeaverTails. A2D’s robust performance on these previously unseen categories suggests that it learns a generalizable refusal mechanism, rather than memorizing dataset-specific patterns. These results, added in **Section 6.4**, provide direct evidence that A2D can handle a broader, more diverse range of harmful content beyond its training set.
>
> * LLaDA Results
>
> **AdvBench**
>
> |Method|Zeroshot ↓|PAIR ↓|ReNe ↓|Perfilling ↓|DIJA ↓|**Avg.** ↓|
> |-|-|-|-|-|-|-|
> |RT|**0.0**|7.5|57.5|82.5|80.0|45.5|
> |SFT|45.0|76.3|63.8|77.5|82.5|69.0|
> |VRPO|0.0|10.0|13.8|1.3|11.3|7.3|
> |A2D|**0.0**|**1.3**|**12.5**|**0.0**|**0.0**|**2.8**|
>
> **JailbreakBench**
>
> |Method|Zeroshot ↓|PAIR ↓|ReNe ↓|Perfilling ↓|DIJA ↓|**Avg.** ↓|
> |-|-|-|-|-|-|-|
> |RT|2.5|20.0|57.5|70.0|61.3|42.3|
> |SFT|51.3|87.5|55.0|72.3|63.8|66.0|
> |VRPO|**1.3**|21.3|16.3|7.5|22.5|13.8|
> |A2D|**1.3**|**7.5**|**15.0**|**1.3**|**1.3**|**5.3**|
>
> * Dream Results
>
> **AdvBench**
>
> |Method|Zeroshot ↓|PAIR ↓|ReNe ↓|Perfilling ↓|DIJA ↓|**Avg.** ↓|
> |-|-|-|-|-|-|-|
> |RT|**0.0**|13.8|51.3|33.8|81.3|36.0|
> |SFT|16.3|46.3|40.0|70.0|85.0|51.5|
> |VRPO|**0.0**|10.0|48.8|7.5|60.0|25.3|
> |A2D|**0.0**|**0.0**|**7.5**|**0.0**|**0.0**|**1.5**|
>
> **JailbreakBench**
> |Method|Zeroshot ↓|PAIR ↓|ReNe ↓|Perfilling ↓|DIJA ↓|**Avg.** ↓|
> |-|-|-|-|-|-|-|
> |RT|**0.0**|16.3|41.3|48.8|67.5|34.8|
> |SFT|27.5|67.5|36.3|68.8|76.3|55.3|
> |VRPO|**0.0**|7.5|38.8|18.8|45.0|22.0|
> |A2D|**0.0**|**6.3**|**8.8**|**0.0**|**0.0**|**3.0**|
>
> [1] Ji, Jiaming, et al. "Beavertails: Towards improved safety alignment of llm via a human-preference dataset." Advances in Neural Information Processing Systems 36 (2023).\
> [2] Zou, Andy, et al. "Universal and transferable adversarial attacks on aligned language models." arXiv preprint arXiv:2307.15043 (2023).\
> [3] Chao, Patrick, et al. "Jailbreakbench: An open robustness benchmark for jailbreaking large language models." Advances in Neural Information Processing Systems 37 (2024).

---

> ### Author Response · Authors · 2025-11-18
> **Rebuttal - Part II**
>
> **3. Scalability of A2D**
>
> > Scalability of Training: The A2D method requires fine-tuning on a dataset of harmful prompts paired with unsafe responses, where harmful spans are identified and masked. It's not clear how this data-generation and training process scales to more complex, "in-the-wild" harmful requests that may not follow clear templates.
>
> We take this opportunity to clarify how A2D operates during training. A2D does not rely on handcrafted harmful spans, explicit span annotations, or curated template structures. Instead, A2D introduces a minimal change to dLLM training. Given a harmful prompt and its corresponding harmful response, a random subset of tokens in the response is masked following the default dLLM training process. The model is then trained to emit $\texttt{[EOS]}$ at all masked positions, thereby learning to suppress harmful outputs at any step, regardless of decoding order. The only change is that masked positions are supervised to emit the $\texttt{[EOS]}$ token rather than reconstruct the original token. This supervision signal is fully automatic, requiring no manual span identification or special preprocessing, and allows the model to learn termination behavior at arbitrary positions.
>
> Regarding data availability, A2D requires only harmful prompt-response pairs and a retain set for utility preservation. Both are readily available in existing safety benchmarks. For instance, WildJailbreak [1] includes 132k harmful requests, and PKU-SafeRLHF [2] provides 265k question-answer pairs annotated with safety meta-labels across 19 harm categories. These large-scale corpora allow A2D to scale naturally without the need for custom data collection or engineering.
>
> [1] Jiang, Liwei, et al. "Wildteaming at scale: From in-the-wild jailbreaks to (adversarially) safer language models." Advances in Neural Information Processing Systems 37 (2024).\
> [2] Ji, Jiaming, et al. "Pku-saferlhf: Towards multi-level safety alignment for llms with human preference." Proceedings of the 63rd Annual Meeting of the Association for Computational Linguistics (2025).

---

> ### Author Response · Authors · 2025-11-18
> **Rebuttal - Part III**
>
> **4. Over-Rejection Risk**
>
> We appreciate the reviewer’s thoughtful question. In practice, we find that A2D handles such prompts reliably. Once a prompt is identified as harmful during decoding, the model emits the $\texttt{[EOS]}$ token at masked positions, effectively halting generation, as demonstrated in XSTest. This behavior prevents harmful content from being completed, while benign prompts remain unaffected.
>
> To further assess the risk of false rejections in benign yet ambiguous cases, particularly in more complex long-form scenarios, we conducted a targeted evaluation using OR-Bench [1]. We curated a subset of the 70 longest prompts that appear superficially harmful but are in fact safe, which tend to be significantly more challenging.
>
> As shown in the table below, A2D achieves the highest compliance rate among all methods on this challenging set. Notably, on LLaDA-1.5, A2D even surpasses the original model in preserving benign completions without premature refusals. These results suggest that A2D’s refusal mechanism remains robust and context sensitive, effectively avoiding unnecessary truncation even under complex and ambiguous input conditions. We have included this result in **Section 6.4**.
>
> * Compliance Rate (↑)
> |Model|Original|RT|VRPO|A2D|
> |-|-|-|-|-|
> |LLaDA-Instruct|55.7|25.7|44.3|47.1|
> |LLaDA-1.5|45.7|28.6|20.0|54.3|
> |Dream|14.3|4.3|12.9|12.9|
>
> [1] Cui, Justin, et al. "Or-bench: An over-refusal benchmark for large language models." 42nd International Conference on Machine Learning (2025).
>
>
> ---
>
> **5. Design of Early Rejection Mechanism**
>
> We deliberately use the $\texttt{[EOS]}$ probability at the leftmost masked position in the first decoding step, as this is the first token the model is guaranteed to generate. The model does not place $\texttt{[EOS]}$ at this position unless it explicitly intends to reject the prompt. In contrast, $\texttt{[EOS]}$ may appear at later positions even in benign completions (e.g., short answers like “okay”), which introduces the risk of false positives. This makes the leftmost token a precise and efficient early rejection signal. While we agree that aggregate heuristics such as mean or max probabilities could offer a more conservative alternative, we found them more prone to misclassifying short safe completions. As shown in Figure 8, this heuristic enables up to 19.3× faster refusals with minimal over-rejection. Additional details are provided in **Section 7**.

---

> > ### Comment · Reviewer_jYvv · 2025-11-25
> >
> > I have read the responses, and I will increase my score to support this work.

---

### Official Review · Reviewer_Mom3 · 2025-10-31

**Soundness:** 3
**Presentation:** 3
**Contribution:** 3
**Rating:** 4
**Confidence:** 4

**Summary:**

This paper introduces a critical safety vulnerability inherent to Diffusion Large Language Models (dLLMs), specifically the fact that their any-order, any-step generation capability greatly expands the attack surface. To address this, the authors propose A2D, a token-level alignment method that trains dLLMs to emit the [EOS] token at masked positions whenever harmful content is encountered. Experiments on three masked dLLMs show A2D outperforms traditional baselines: it reduces attack success rates, preserves core capabilities, and improves monitoring efficiency.

**Strengths:**

- The paper introduces the first token-level safety alignment method tailored to dLLMs. The proposed A2D method is simple, effective, and addresses two unique vulnerabilities—any-order decoding and any-step adversarial attacks.
- The method strikes a balance between safety and general utility.
- The real-time [EOS] probability monitoring and early rejection mechanism introduces a measurable efficiency benefit.

**Weaknesses:**

- Lack of Generalization Discussion: The paper focuses exclusively on masked diffusion dLLMs (LLaDA, LLaDA-1.5, Dream). While this is a prominent class, it would strengthen the paper to discuss the theoretical or practical challenges of extending A2D to other LLM architectures.
- Potential for Refusal Token Attack: The use of the single [EOS] token as the universal suppression signal is clever but introduces a single point of failure. A minor discussion on the robustness of the [EOS] signal itself—i.e., the risk of an adversary explicitly targeting or corrupting the refusal behavior of this specific token—would complete the safety analysis.
- Inadequate Assessment: The main experiment is only conducted on the Harmbench, making the assessment of A2D’s generalization to the full spectrum of real-world threats insufficient.
- Limited Practical Value: With few existing landing scenarios for dLLMs, A2D’s practical impact is currently constrained.

**Questions:**

- Could A2D be adapted to other LLM architectures?
- A2D relies on [EOS] as the refusal token, but attackers might attempt to induce dLLMs to reduce the generation probability of the targeted token. Have the authors tested A2D’s robustness against such [EOS]-interference attacks?

---

> ### Author Response · Authors · 2025-11-18
> **Rebuttal - Part I**
>
> Thank you for the reviewer’s thoughtful and detailed comments. The points raised about generalization, refusal robustness, scalability, and practical deployment helped us refine and clarify several central aspects of A2D. We hope our response addresses the concerns you raised.
>
> ---
>
> **1. Applicability of A2D**
>
> Thank you for raising this important point. Our focus on diffusion LLMs (dLLMs) stems from a structural safety vulnerability unique to their architecture. Unlike autoregressive models that decode strictly left-to-right, diffusion LLMs generate in an any-order, any-step manner. This order-agnostic decoding allows harmful tokens to appear at arbitrary positions and enables adversaries to instantiate harmful templates through partially prefilled contexts. A2D is therefore designed specifically to address this dLLM-specific weakness by aligning safety at the precise moment a harmful continuation appears, regardless of position or decoding step.
>
> The core idea of A2D, however, is not tied to the diffusion architecture itself. Its underlying principle is to train the model to emit a terminating token ($\texttt{[EOS]}$) when conditioned on harmful text, and this approach is inherently architecture-agnostic and can be applied to autoregressive models as well. To explore this, we conducted preliminary experiments adapting A2D to two non-diffusion models, Qwen-2.5-7B and LLaMA-3.1-8B. In this setting, we conditioned the model on a harmful prefix and trained subsequent positions to predict a special terminating token $\texttt{[EOS]}$, applying the loss only to this position. For example,
>
> > $\texttt{Q: harmful prompt → A: Okay, here is [EOS]}$
>
> As shown below, Even with this minimal adaptation, both models show clear reductions in jailbreak attack success rates under strong attacks such as Prefilling, PAIR, and ReNe.
>
> * Result on Qwen-2.5
> |Model|ZerosShot ↓|PAIR ↓|ReNe ↓|Prefilling ↓|Avg. ↓|
> |-|-|-|-|-|-|
> |Qwen|2.5|11.9|47.5|47.5|27.4|
> |Qwen+A2D|0.0|11.3|5.0|0.0|4.1|
>
> * Result on LLaMA-3.1
> |Model|ZerosShot ↓|PAIR ↓|ReNe ↓|Prefilling ↓|Avg. ↓|
> |-|-|-|-|-|-|
> |LLaMA|5.0|42.5|41.3|46.3|33.8|
> |LLaMA+A2D|0.0|29.4|30.6|0.0|15.0|
>
> While a full generalization study is beyond the scope of the current work, our preliminary findings suggest that the core alignment mechanism of A2D extends beyond diffusion LLMs and may provide a general framework for suppressing harmful continuations across architectures. We have added these expanded results in **Appendix C.5**.

---

> ### Author Response · Authors · 2025-11-18
> **Rebuttal - Part II**
>
> **2. A2D against [EOS]-interference attack**
>
> Thank you for raising this consideration. While A2D uses $\texttt{[EOS]}$ as the universal suppression signal, we view this as a practical design choice rather than an inherent point of fragility. To directly assess whether an adversary could suppress or override the $\texttt{[EOS]}$ signal, we designed two explicit EOS-targeting attacks. The first, **Long-Answer**, prompts the model to generate unusually long continuations, counteracting its tendency to terminate. The second, **Do-Not-EOS**, explicitly instructs the model not to output $\texttt{[EOS]}$. Both are crafted to weaken or eliminate the refusal behavior by directly suppressing the $\texttt{[EOS]}$ token.
>
> We combined these two alternatives to the ReNe jailbreak attack [1]. Even under these EOS-suppression attacks, A2D continues to achieve low attack success rates, demonstrating that its defensive capability holds up even when adversaries explicitly target the $\texttt{[EOS]}$ signal. As shown in the table below, although alternative refusal mechanisms may be worth exploring, our experiments indicate that A2D remains robust despite direct attempts to suppress its $\texttt{[EOS]}$-based alignment signal. We have included this result in **Appendix C.6**.
>
> * Attack Success Rates of ReNe Variants Aligned with A2D
>
> |Model|Original|ReNe ↓|ReNe (Long-Answer) ↓|ReNe (Do-Not-EOS) ↓|
> |-|-|-|-|-|
> |LLaDA|56.3|17.5|13.1|12.5|
> |Dream|40.6|9.4|3.8|1.9|
>
> [1] Ding, Peng, et al. "A wolf in sheep’s clothing: Generalized nested jailbreak prompts can fool large language models easily." Proceedings of the 2024 Conference of the North American Chapter of the Association for Computational Linguistics (2024).
>
> ---
>
> **3. Broader Assessment**
>
> A key finding of our study is that A2D generalizes well beyond its training distribution. Although the model is tuned only on BeaverTails [1], it already shows strong performance on HarmBench [2]. Following the reviewer’s suggestion, we additionally evaluated A2D on AdvBench [3] and JailbreakBench [4], which include threat patterns and adversarial structures not present in BeaverTails. Across both datasets, A2D maintains low attack success rates and consistently outperforms other finetuning-based methods by a substantial margin. These results suggest that A2D captures transferable safety signals rather than overfitting to a narrow set of examples. The expanded evaluation is now included in **Section 6.4**.
>
> * Results on LLaDA
>
> **AdvBench**
>
> |Method|Zeroshot ↓|PAIR ↓|ReNe ↓|Perfilling ↓|DIJA ↓|**Avg.** ↓|
> |-|-|-|-|-|-|-|
> |RT|**0.0**|7.5|57.5|82.5|80.0|45.5|
> |SFT|45.0|76.3|63.8|77.5|82.5|69.0|
> |VRPO|0.0|10.0|13.8|1.3|11.3|7.3|
> |A2D|**0.0**|**1.3**|**12.5**|**0.0**|**0.0**|**2.8**|
>
> **JailbreakBench**
>
> |Method|Zeroshot ↓|PAIR ↓|ReNe ↓|Perfilling ↓|DIJA ↓|**Avg.** ↓|
> |-|-|-|-|-|-|-|
> |RT|2.5|20.0|57.5|70.0|61.3|42.3|
> |SFT|51.3|87.5|55.0|72.3|63.8|66.0|
> |VRPO|**1.3**|21.3|16.3|7.5|22.5|13.8|
> |A2D|**1.3**|**7.5**|**15.0**|**1.3**|**1.3**|**5.3**|
>
> * Results on Dream
>
> **AdvBench**
>
> |Method|Zeroshot ↓|PAIR ↓|ReNe ↓|Perfilling ↓|DIJA ↓|**Avg.** ↓|
> |-|-|-|-|-|-|-|
> |RT|**0.0**|13.8|51.3|33.8|81.3|36.0|
> |SFT|16.3|46.3|40.0|70.0|85.0|51.5|
> |VRPO|**0.0**|10.0|48.8|7.5|60.0|25.3|
> |A2D|**0.0**|**0.0**|**7.5**|**0.0**|**0.0**|**1.5**|
>
> **JailbreakBench**
>
> |Method|Zeroshot ↓|PAIR ↓|ReNe ↓|Perfilling ↓|DIJA ↓|**Avg.** ↓|
> |-|-|-|-|-|-|-|
> |RT|**0.0**|16.3|41.3|48.8|67.5|34.8|
> |SFT|27.5|67.5|36.3|68.8|76.3|55.3|
> |VRPO|**0.0**|7.5|38.8|18.8|45.0|22.0|
> |A2D|**0.0**|**6.3**|**8.8**|**0.0**|**0.0**|**3.0**|
>
> [1] Ji, Jiaming, et al. "Beavertails: Towards improved safety alignment of llm via a human-preference dataset." Advances in Neural Information Processing Systems 36 (2023).\
> [2] Mazeika, Mantas, et al. "Harmbench: A standardized evaluation framework for automated red teaming and robust refusal." 41st International Conference on Machine Learning (2024).\
> [3] Zou, Andy, et al. "Universal and transferable adversarial attacks on aligned language models." arXiv preprint arXiv:2307.15043 (2023).\
> [4] Chao, Patrick, et al. "Jailbreakbench: An open robustness benchmark for jailbreaking large language models." Advances in Neural Information Processing Systems 37 (2024).

---

> ### Author Response · Authors · 2025-11-18
> **Rebuttal - Part III**
>
> **4. Practical Impact of A2D**
>
> Thank you for raising these important and insightful points. While diffusion LLMs (dLLMs) are still in the early stages of deployment, their practical relevance is already emerging. Mercury [1] is publicly available, and models such as LLaDA [2] and Dream [3] achieve performance comparable to LLaMA-sized autoregressive models despite being early-stage systems. These developments indicate that dLLMs are a promising direction for next-generation LLM design, offering advantages such as flexible any-order decoding, controllable denoising trajectories, and parallel token refinement.
>
> The vulnerability that A2D targets arises directly from this any-order, any-step diffusion decoding process rather than from the specifics of current models. Order-agnostic generation introduces safety failure modes, such as prefilling-based attacks [4], that do not occur in conventional autoregressive architectures. Addressing these risks early is important for ensuring that the growing class of diffusion-based language models is supported by a principled safety foundation.
>
> We therefore view A2D not as a response to transient limitations of current dLLMs, but as an early mitigation of a structural safety risk inherent to diffusion-based generation, a risk that becomes increasingly relevant as these models mature and see wider adoption.
>
> [1] Khanna, Samar, et al. "Mercury: Ultra-fast language models based on diffusion." arXiv preprint arXiv:2506.17298 (2025).\
> [2] Nie, Shen, et al. "Large language diffusion models." Advances in Neural Information Processing Systems 38 (2025).\
> [3] Ye, Jiacheng, et al. "Dream 7b: Diffusion large language models." arXiv preprint arXiv:2508.15487 (2025).\
> [4] Wen, Zichen, et al. "The devil behind the mask: An emergent safety vulnerability of diffusion llms." arXiv preprint arXiv:2507.11097 (2025).

---

### Official Review · Reviewer_dkrr · 2025-11-01

**Soundness:** 4
**Presentation:** 3
**Contribution:** 3
**Rating:** 8
**Confidence:** 5

**Summary:**

Recently, the safety alignment of diffusion LLMs (dLLMs) was shown to be easily circumvented by a decoding exploit called DiJA, a generalization of the prefilling attack against autoregressive LLMs to dLLMs. This work proposes a simple fine-tuning defense against DiJA called A2D, where the model learns to output EOS tokens when encountering a DiJA attack, rather than harmful content. Through extensive experiments, A2D is shown to significantly reduce the success of DiJA across multiple models, safety benchmarks and decoding strategies. Various other attacks (e.g., PAIR) are also evaluated, and their success is also shown to significantly degrade under A2D.

**Strengths:**

1. A KL-divergence-based study to explore the shallowness of the safety alignment of dLLMs is conducted, following the approach of [1]. The study reveals a novel insight specific to dLLMs: that shallow safety aligmment is exhibited under differing orders of mask token denoising (“any order”).
2. The design of the A2D defense is simple to implement and intuitive. Experimental results show that A2D is highly effective at reducing the success of DiJA and prefilling attacks across various decoding strategies, and a followup KL-divergence study shows that deep safety alignment has been effectively learned both in terms of decoding step and order.
3. A “worst-case” version of DiJA called FITS is evaluated to test the scenario where everything in the assistant response except one (masked) sentence contains harmful content. Even under this challenging scenario, A2D effectively reduces the attack success to near zero.

[1] Qi, Xiangyu, et al. "Safety alignment should be made more than just a few tokens deep." arXiv preprint arXiv:2406.05946 (2024).

**Weaknesses:**

1. During evaluation, there are two types of randomness at play: randomness from sampling from the model’s output probability distribution, and randomness in the decoding strategy. Hence, it would be good to evaluate over multiple runs and report the mean and standard devations.
2. Compliance on XSTest is reported in Figure 5b, yet compliance does not necessarily indicate the outputs are useful under the challenging XSTest setting. Can some evaluation of response quality also be reported for XSTest?
3. Notational discrepancies:
- Line 152-153: “For each selected index i” is mentioned, but then “for all i” is mentioned in the equation. The equation’s “for all i” statement should be modified to indicate only the selected subset of mask tokens.

**Questions:**

1. Are the data points plotted in Figure 3 averages across the 150 prompts? It would be good to also depict a measure of spread (e.g., standard deviation).
2. In Table 1, can results for the original models (i.e., without additional fine-tuning) also be reported?

---

> ### Author Response · Authors · 2025-11-18
> **Rebuttal**
>
> We thank the reviewer for the positive assessment. We appreciate that the reviewer accurately captured the key points we intended to convey regarding shallow safety alignment in dLLMs and the effectiveness of A2D in achieving deeper alignment across any-step and any-order. We hope the following response addresses the reviewer’s remaining points.
>
> ---
>
> **1. Improved Evaluation Completeness and Reproducibility**
>
> We agree that randomness in both sampling and decoding should be properly accounted for. We therefore evaluated the main results using three independent random seeds and have added mean and standard deviations to Table 1 (main results) and Table 12 (detailed results). In addition, we have added the result of the original model (i.e., before any fine-tuning) to Table 1 for completeness. Note that we do not report a standard deviation for the aggregated “average” score, because this value is a deterministic arithmetic mean of different benchmark metrics rather than a quantity obtained from repeated runs.
> Additionally, regarding the question below:
>
> > Are the data points plotted in Figure 3 averages across the 150 prompts? It would be good to also depict a measure of spread (e.g., standard deviation).
>
> Yes, Figure 3 reports the average over 150 prompts from the BeaverTails [1]. Following the reviewer’s suggestion, we have updated Figure 3 to include the corresponding standard deviations. For completeness, the same update has been applied to Figure 4.
>
> [1] Ji, Jiaming, et al. "Beavertails: Towards improved safety alignment of llm via a human-preference dataset." Advances in Neural Information Processing Systems 36 (2023).
>
> ---
>
> **2. Evaluation Response Quality for XSTest**
>
> We appreciate the reviewer for raising this important point regarding the response quality of outputs under XSTest. The reviewer is correct that high compliance does not automatically ensure that the generated responses remain useful or well formed, especially under the challenging conditions of XSTest. To directly assess response quality, we evaluated the utility of each output on a 1-10 scale using the LLM-as-a-Judge setup [1] based on GPT-4o.  As shown in the table below, response quality decreases only slightly while still remaining stronger than other methods, indicating that the model preserves practical response quality even under XSTest.
>
> * Results on Response Quality (↑)
> ||Original|RT|VRPO|A2D|
> |-|-|-|-|-|
> |Response quality|6.8|4.4|6.4|6.5|
>
> [1] Zheng, Lianmin, et al. "Judging llm-as-a-judge with mt-bench and chatbot arena." Advances in neural information processing systems 36 (2023).
>
> ---
>
> **3. Correcting Notation for Masked Token Selection**
>
> We thank the reviewer for pointing out this inconsistency. The original “for all i” incorrectly implied that the equation applied to every position, while the preceding explanation referred only to the selected mask-token indices. We fixed the notation accordingly **(Line 154-156)**.

---

### Author Response · Authors · 2025-12-03
**Summary of Rebuttal**

We sincerely thank the reviewers and the Area Chair for their constructive feedback and the time dedicated to reviewing our work. We are encouraged that the reviewers highlighted our work’s key strengths: the novelty and significance of identifying dLLM-specific vulnerabilities and proposing the first tailored token-level defense (**dkrr, Mom3, jYvv, SJBR**), the method’s simplicity and effectiveness (**dkrr, Mom3, jYvv**), its robustness against strong attacks like DIJA and FITS (**dkrr, jYvv, SJBR**), and its ability to balance safety with utility and efficiency (**Mom3, SJBR**). We are also grateful to Reviewer **jYvv**, who found our rebuttal convincing and expressed support for the paper.

Below, we summarize how we have addressed the key concerns raised during the review process, supported by new experiments and analyses.

---

**1. Demonstrating Generalization Beyond Training Data (Reviewers Mom3, jYvv, SJBR)**

Concerns were raised regarding the reliance on BeaverTails. To demonstrate that A2D does not overfit to a specific dataset, we expanded our evaluation to include AdvBench and JailbreakBench. A2D achieved consistently superior performance across these diverse benchmarks, proving its effectiveness against threat patterns and adversarial structures not present in the training set.

**2. Verifying Robustness Against Adaptive Attacks (Reviewer Mom3)**

To address concerns that the [EOS] token acts as a single point of failure, we evaluated A2D against adaptive [EOS]-interference attacks (specifically Long-Answer and Do-Not-EOS variants). Our results confirm that A2D remains robust even when adversaries explicitly attempt to suppress the refusal signal, maintaining low attack success rates where baselines fail.

**3. Demonstrating Robustness Against Over-Refusal (Reviewers jYvv, SJBR)**

We addressed concerns regarding potential over-refusal on complex benign queries. On the challenging OR-Bench (long-form benign prompts that appear harmful), A2D achieved the highest compliance rate, outperforming both baselines and the original model in avoiding false refusals. This confirms that A2D’s refusal mechanism is highly precise and does not compromise the model’s ability to handle ambiguous benign contexts.

**4. Extending Applicability to Autoregressive Architectures (Reviewer Mom3)**

While our primary focus is the unique vulnerability of diffusion LLMs, we addressed the question of broader applicability. We applied the core A2D principle to autoregressive models (Qwen-2.5 and LLaMA-3.1). Preliminary results show that A2D effectively reduces jailbreak success rates in these architectures as well, suggesting the method’s potential as a generalizable safety framework.

**5. Clarifying Computational Efficiency (Reviewer SJBR)**

We provided a detailed cost analysis confirming that A2D incurs no additional computational overhead compared to standard SFT during training or inference. Furthermore, we highlighted that the [EOS]-based early rejection mechanism improves inference efficiency by up to 19.3$\times$ for harmful queries.

---

We believe the revised manuscript and additional experiments fully resolve the reviewers’ concerns and demonstrate the significance and robustness of A2D. We hope our contributions will help advance the foundational safety of emerging diffusion large language models and pave the way for more robust alignment in non-autoregressive generation.

---

### Meta-Review · Area_Chair_LZrc · 2025-12-31

**Summary:**

This paper introduces A2D, a token-level safety alignment method designed specifically for diffusion large language models (dLLMs). dLLMs’ any-order, any-step generation flexibility expands the attack surface, making them vulnerable to template-based prefilling attacks (e.g., DIJA). The paper identifies a clear and novel safety vulnerability in diffusion LLMs and proposes a simple, effective, and theoretically grounded defense. The rebuttal comprehensively addressed the majority of reviewer concerns, with additional experiments strengthening the claims on generalization, robustness, and practicality.

**Reviewer Concerns:**

Reviewer dkrr requests for reproducibility (multi-run statistics, standard deviations) and response quality evaluation on XSTest were met in the rebuttal. Reviewer Mom3 concerns about the limited discussion of practical impact given early-stage dLLM deployment. For Reviewer jYvv and Reviewer SJBR, minor questions about threshold calibration across models remain open but do not affect core contribution.

**Reviewer Scores:**

# dkrr:
 Initially 8 (Accept). Likely unchanged; concerns were addressed.

# Mom3:
Initially 4 (Marginally Below Threshold). Likely raised to 6 given thorough responses to generalization and robustness concerns.

# jYvv:
Initially 4 (Marginally Below Threshold). Explicitly increased score after rebuttal.

# SJBR:
Initially 6 (Marginally Above Threshold). Likely unchanged.

---

### Decision · Program_Chairs · 2026-01-26

Accept (Poster)